# Model-Based Reinforcement Learning under Random Observation Delays

## Abstract

Delays frequently occur in real-world environments, yet standard reinforcement learning (RL) algorithms often assume instantaneous perception of the environment. We study random sensor delays in POMDPs, where observations may arrive out-of-sequence, a setting that has not been previously addressed in RL. We analyze the structure of such delays and demonstrate that naive approaches, such as stacking past observations, are insufficient for reliable performance. To address this, we propose a model-based filtering process that sequentially updates the belief state based on an incoming stream of observations. We then introduce a simple delay-aware framework that incorporates this idea into model-based RL, enabling agents to effectively handle random delays. Applying this framework to Dreamer, we compare our approach to delay-aware baselines developed for MDPs. Our method consistently outperforms these baselines and demonstrates robustness to delay distribution shifts during deployment. Additionally, we present experiments on simulated robotic tasks, comparing our method to common practical heuristics and emphasizing the importance of explicitly modeling observation delays.

## 1 Introduction

Despite the remarkable success of reinforcement learning (RL) across a wide range of domains, standard RL algorithms rely on the assumption of delay-free interaction with the environment. In practice, however, delays are pervasive and often unavoidable, particularly in real-world systems such as robotics, autonomous driving, and distributed control (Abadía et al., 2021; Mahmood et al., 2018; Fagundes-Junior et al., 2023). These delays can occur at different stages of the pipeline, such as sensing, processing, and communication. They are generally divided into two types: (i) *feedback delays*, the time lag in receiving observations, and (ii) *execution delays*, describing the delay between selecting an action and its execution in the environment. While these are highly common in practical applications, they are typically ignored or oversimplified in the RL literature.

When delays are present, a common workaround in robotics is to issue "no-op" actions, effectively instructing the agent to wait until the delayed observation arrives (Walsh et al., 2007). However, this approach is often impractical or even unsafe. For example, an autonomous vehicle detecting an unexpected approaching obstacle in a low-latency sensor cannot afford to wait for its other sensors before acting, as doing so may result in a collision. Even defaulting to braking may not be viable if the obstacle is too close, and the vehicle may need to infer the safest swerving motion under the uncertainty caused by delayed perception.

Even when delays are explicitly considered, they are often addressed with simplifying assumptions that fail to capture their full complexity in real-world tasks. Existing approaches assume either a fully observable environment, as in Markov Decision Processes (MDPs) (Katsikopoulos & Engelbrecht, 2003; Liotet et al., 2021; Derman et al., 2021; Liotet et al., 2022; Wu et al., 2024a), or fixed delays in Partially Observable MDPs (POMDPs) (Kim & Jeong, 1987; Karamzade et al., 2024). However, real-world systems often involve partial observability and random delays. Unlike MDPs, where a single observation fully represents the environment's state, POMDPs require the agent to integrate past observations to maintain a belief over the state. With random observation delays, observations may arrive out-of-sequence (OOS), a phenomenon that does not arise in fixed-delay settings and is innocuous in MDPs, where the most recent observation suffices for decision-making. In contrast,

POMDPs require reasoning over past observations, and under random delays, relying solely on the most recent observation is insufficient for optimal control.

In this work, we consider random observation delays in POMDPs. To address the OOS phenomenon, we propose a latent-space filtering approach that enables effective learning in the presence of OOS observations imposed by random delays. By leveraging model-based approaches, our method forms a belief over the current latent state, given the set of available observations. In particular, the filtering process exploits a world model trained in the delayed environment to sequentially update the belief based on received observations. This belief state then serves as a sufficient statistic for policy learning under an incomplete set of observations to ensure actions are informed solely by available inputs.

We conduct a series of experiments on both synthetic control tasks and realistic simulated robotic environments. Our method not only outperforms existing approaches in fully observable MDP settings but is also the only one capable of handling more realistic, partially observable scenarios when facing longer delays. Notably, our approach demonstrates strong generalization to test-time delay distributions compared to baseline methods: when trained on a wider delay distribution, it performs significantly better under shorter test-time delays and shows minimal performance degradation under longer ones. These results highlight the potential of our method for real-world deployment, where delay patterns are often unknown in advance and potentially nonstationary.

Here, we summarize the contributions of this paper as follows. (i) We study random observation delays in POMDPs and propose a framework that connects this setting to standard POMDP formulations. (ii) We introduce a filtering procedure for processing OOS observations within model-based RL. (iii) We present the first method designed for this setting and describe how to integrate the filtering process into existing model-based RL algorithms. (iv) We conduct extensive experiments across diverse environments, demonstrating superior performance over baselines and strong generalization to unseen delay distributions.

## 2 PRELIMINARIES

A *Partially Observable Markov Decision Process (POMDP)* is a tuple $\mathcal{M} = \langle S, A, \mathcal{T}, r, \Omega, O, \gamma \rangle$, where $S$, $A$, and $\Omega$ denote the sets of states, actions, and observations, respectively. The transition dynamics are defined by $\mathcal{T}(s' \mid s, a)$, the reward function by $r(s, a)$, and the observation (emission) probabilities by $O(o \mid s)$. At each timestep $t$, the environment is in state $s_t \in S$, the agent receives an observation $o_t \sim O(o_t \mid s_t)$, selects an action $a_t \in A$, receives reward $r_t = r(s_t, a_t)$, and transitions according to $s_{t+1} \sim \mathcal{T}(s_{t+1} \mid s_t, a_t)$. The objective is to select actions that maximize the expected return $\mathbb{E}[\sum_{t=0}^{\infty} \gamma^t r_t]$, where $\gamma \in [0, 1)$ is the discount factor.

### 2.1 MODEL-BASED RL

Recent model-based RL approaches focus on learning a latent dynamics model, or world model, that captures the environment's behavior and enables long-term prediction (Ha & Schmidhuber, 2018; Hansen et al., 2022; Micheli et al., 2022; Hafner et al., 2025). In this framework, the agent maintains a latent state $x_t$ governed by a parametrized transition model $p_\theta(x_t \mid x_{t-1}, a_{t-1})$, and generates observations through a decoder $p_\theta(o_t \mid x_t)$. Note that rewards are usually part of the model, but here we omit them for brevity. Since the latent state is not directly observable in training data, a variational posterior $q_\theta(x_{1:T} \mid o_{1:T}, a_{1:T}) = \prod_t q_\theta(x_t \mid x_{t-1}, o_t, a_{t-1})$, first proposed by Hafner et al. (2019), can be used for the distribution of the latent state sequence of a particular observed episode $(o_{1:T}, a_{1:T})$ of length $T$. The model is trained by maximizing an evidence lower bound (ELBO) on the sequence log-likelihood, leading to the objective (Hafner et al., 2019):

$$\mathcal{L} = \sum_{t=1}^{T} \mathbb{E}_{q_\theta}[\ln p_\theta(o_t \mid x_t)] - \mathbb{E}_{q_\theta}[\mathbb{D}[q_\theta(x_t \mid x_{t-1}, o_t, a_{t-1}) \,\|\, p_\theta(x_t \mid x_{t-1}, a_{t-1})]], \quad (1)$$

where $\mathbb{D}$ is the Kullback–Leibler (KL) divergence. This objective encourages the latent state to retain sufficient information for reconstructing observations, while remaining consistent with the prior dynamics $p_\theta$. Throughout the text, we omit the dependence on $\theta$ whenever it is clear from context.

Many existing works exploit this or similar models for reinforcement learning and planning (Ha & Schmidhuber, 2018; Hafner et al., 2019; Micheli et al., 2022; Zhang et al., 2023). One notable

example is Dreamer (Hafner et al., 2025), which trains policies entirely within a learned Recurrent State Space Model (RSSM) world model. Dreamer alternates between three key stages: (1) training the world model, (2) learning the policy through imagined trajectories, and (3) collecting new experiences.

## 3  RANDOM OBSERVATION DELAY ENVIRONMENTS

We define a *Random Observation Delay Environment* as a pair $\langle \mathcal{M}, \mathcal{D} \rangle$, where $\mathcal{M}$ is a standard POMDP and $\mathcal{D}$ is a distribution over non-negative integers representing stochastic delays. At each time step $t$, the observation generated by the environment is not revealed immediately, but is instead delivered after a random delay $d_t \sim \mathcal{D}^1$. That is, $o_t$ becomes available at time $t + d_t$. The agent's actual observation at time $t$ consists of all information scheduled to arrive at that step. We denote this (possibly empty) set by $\tilde{o}_t = \{(o_\tau, \tau) : \tau + d_\tau = t\}$. We assume the agent observes the timestamp of delivered observations, but not $d_\tau$ for undelivered observations. We further assume that all delays are finite, with $D$ denoting the maximum possible delay.

In the fully observable setting, where $O(o_t = s_t \mid s_t) = 1$, an equivalent delay-free MDP can be constructed by augmenting the state with the sequence of actions taken since the most recently observed state (Katsikopoulos & Engelbrecht, 2003). Similarly, in partially observable environments with constant delays, one can construct an equivalent delay-free POMDP by augmenting the state, while preserving the original observation space (Karamzade et al., 2024). In both cases, the augmentation only needs to track a sequence of past actions. However, with random delays, this structure is no longer sufficient, and reducing a delayed environment to a standard POMDP becomes more complex.

**Reducing to a standard POMDP**  Under random observation delays, the agent may potentially receive any nonnegative number of observations at each timestep. The new observation space becomes $\widetilde{\Omega} = \bigcup_{k=0}^{D+1} (\Omega \times \mathbb{N})^k$, where each new observation $\tilde{o}_t$ is a set of observations delivered at time $t$, each paired with its original emission timestamp. To encode this within a standard POMDP, we augment the state with a buffer $u_t$ that stores all observations not previously received:

$$u_t = \{(o_\tau, \tau, d_\tau) : t - d_\tau \leq \tau \leq t\} \in U,$$

where $U$ is the space of sets of observation–timestamp–delay tuples. The augmented state space is $\widetilde{S} = S \times U$, and the equivalent delay-free POMDP is given by the tuple $\langle \widetilde{S}, A, \widetilde{\mathcal{T}}, \widetilde{r}, \widetilde{\Omega}, \widetilde{O}, \gamma \rangle$, with:

$$\widetilde{r}((s_t, u_t), a_t) = r(s_t, a_t),$$
$$\widetilde{O}(\tilde{o}_t \mid (s_t, u_t)) = \delta(\tilde{o}_t = \{(o_\tau, \tau) : (o_\tau, \tau, t - \tau) \in u_t\}),$$
$$\widetilde{\mathcal{T}}((s_{t+1}, u_{t+1}) \mid (s_t, u_t), a_t) = \mathcal{T}(s_{t+1} \mid s_t, a_t) \cdot \mathbb{P}_\mathcal{D}(u_{t+1} | u_t, s_{t+1})$$

where the observation function deterministically returns all entries in the buffer that are scheduled for delivery at time $t$, and each new observation is drawn and stored in the buffer with its associated delay

$$\mathbb{P}_\mathcal{D}(u_{t+1} | u_t, s_{t+1}) = O(o_{t+1} | s_{t+1}) \mathcal{D}(d_{t+1}),$$

when

$$u_{t+1} = \{(o_\tau, \tau, d_\tau) \in u_t : \tau + d_\tau > t\} \cup \{(o_{t+1}, t + 1, d_{t+1})\}$$

for any $o_{t+1}$ and $d_{t+1}$, and assigning probability 0 to any $u_{t+1}$ that does not have this form.

This construction enables the use of standard POMDP algorithms, but at the cost of exponentially increased state and observation space sizes. In particular, the agent's belief must capture uncertainty not only over the latent state but also over pending, undelivered observations. This added complexity is fundamental to POMDPs with random delays, as observations may arrive OOS and cannot be ignored or replaced by the most recent one, unlike in MDPs.

---

[1]Here we assume delays are i.i.d for simplicity. However, our analysis holds for non-stationary or state-dependent delays with some adjustments.

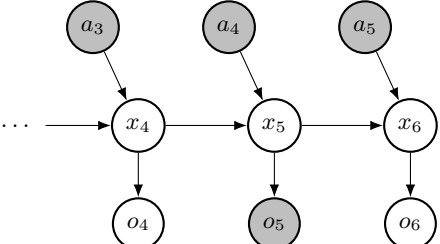

Figure 1: Graphical model of a POMDP with random observation delays at time $t = 6$, when $o_{1:3}$ have already been received before, $o_5$ has just arrived, and $o_4$ and $o_6$ are still pending. Here we have $\tilde{o}_t = \{(o_5, 5)\}$, $\bar{o}_{1:t}$ containing all pairs up to time $t$ except $\{(o_4, 4), (o_6, 6)\}$, and $\kappa_t = 3$.

**World Model for the Reduced POMDPs** In the reduced POMDP formulation, the latent state at time $t$ must be inferred from the set of observations received up to that point. Therefore, $o_t$ would be replaced by $\tilde{o}_t$ in Eq. (1) and the resulting ELBO becomes:

$$\sum_{t=1}^{T} \mathbb{E}[\ln p(\tilde{o}_t \mid x_t)] - \mathbb{E}[\mathbb{D}(q(x_t \mid x_{t-1}, \tilde{o}_t, a_{1:t}) \,\|\, p(x_t \mid x_{t-1}, a_{t-1}))]. \tag{2}$$

While this formulation enables the use of standard model-based RL tools in the delayed setting, it treats the set of received observations as a generic input and does not explicitly model the delay process. As a result, the model may learn unnecessarily complex dynamics to compensate for the partial and OOS nature of the observations, instead of exploiting the structure imposed by the delays.

## 4 DELAY-AWARE MODEL-BASED RL

We introduce a latent-space filtering approach to address random observation delays and OOS inputs within the context of model-based reinforcement learning (MBRL). By maintaining a belief over the current latent state using only the subset of received observations, the agent can make informed decisions despite incomplete information. This section presents the belief update formulation and outlines how it is incorporated into the MBRL framework.

### 4.1 OUT-OF-SEQUENCE FILTERING VIA A WORLD MODEL

Let $\bar{o}_{t_1:t_2} = \bigcup_{t_1 \le \tau \le t_2} \tilde{o}_\tau$ denote the set of all observation and timestamp pairs received between $t_1$ and $t_2$, inclusively. The information available to the agent at time $t$ is $\bar{o}_{1:t}$ and the entire sequence of past actions $a_{1:t-1}$. The objective is to compute the filtered belief $\phi_t$ over the latent state $x_t$, conditioned on the current knowledge, defined as the posterior $\phi_t := p(x_t \mid \bar{o}_{1:t}, a_{1:t-1})$.

We define an auxiliary transition distribution $\psi$ over latent states that retroactively incorporates information available at time $t$ into the filtering process using the learned models $q_\theta$ and $p_\theta$:

$$\psi_\theta(x_\tau \mid x_{\tau-1}, \bar{o}_{1:t}, a_{\tau-1}) = \begin{cases} q_\theta(x_\tau \mid x_{\tau-1}, o_\tau, a_{\tau-1}) & \text{if } (o_\tau, \tau) \in \bar{o}_{1:t}, \\ p_\theta(x_\tau \mid x_{\tau-1}, a_{\tau-1}) & \text{otherwise.} \end{cases}$$

This auxiliary kernel $\psi$ serves as a time-dependent transition function that updates the state based on whether an observation at time $\tau$ is available at time $t$. It uses the variational posterior when $o_\tau$ is observed, and otherwise defaults to the prior dynamics model.

Let $\kappa_t$ denote the most recent timestamp for which $\bar{o}_{1:t}$ includes all observations up to that point, i.e., $\{(o_\tau, \tau)\}_{\tau=1,\ldots,\kappa_t} \subseteq \bar{o}_{1:t}$ (see Fig. 1). Then we can write $\phi_t$ as

$$\phi_t = p(x_t \mid o_{1:\kappa_t}, a_{1:\kappa_t-1}, \bar{o}_{\kappa_t+1:t}, a_{\kappa_t:t-1}) \tag{3}$$

$$= \mathbb{E}_{x_{\kappa_t} \sim q(\cdot \mid x_{\kappa_t-1}, o_{\kappa_t}, a_{\kappa_t-1})}[p(x_t \mid x_{\kappa_t}, \bar{o}_{\kappa_t+1:t}, a_{\kappa_t:t-1})] \tag{4}$$

$$= \mathbb{E}_{x_{\kappa_t} \sim q} \mathbb{E}_{x_{\kappa_t+1} \sim \psi} \ldots \mathbb{E}_{x_{t-1} \sim \psi}[\psi(x_t \mid x_{t-1}, \bar{o}_{\kappa_t+1:t}, a_{t-1})]. \tag{5}$$

---

**Algorithm 1** Delay-Aware MBRL

---

**Require:** $\mathcal{A}$: Model-Based RL algorithm optimizing objective equation 1
    ▷ Inference Mode
1: Initialize obs_buffer
2: **for** time $t$ in episode **do**
3:     Receive $\tilde{o}_t$ and update obs_buffer
4:     Compute $\phi_t$ using Eq. (5) with $p$, $q$, and obs_buffer
5:     Execute action $a_t \sim \pi(\cdot \mid \phi_t)$

    ▷ Training Phase
1: **for** each update step **do**
2:     Collect data with $\pi$ using inference mode and store it in replay buffer $B$
3:     Sample $(o_{1:N}, a_{1:N}, r_{1:N}, \mathcal{J}_{1:N}) \sim B$
4:     Update world model $(p, q)$ with sampled data according to $\mathcal{A}^2$
5:     Compute beliefs $\phi_{1:N}$ using Eq. (5)
6:     Update $\pi(\cdot \mid \phi_t)$ (and $V(\phi_t)$ if applicable) using $\mathcal{A}$

---

We start by splitting the available observations and actions at time $\kappa_t$: this gives us a prefix of received observations $\{o_{1:\kappa_t}, a_{1:\kappa_t-1}\}$, and the remaining delayed ones $\{\bar{o}_{\kappa_t+1:t}, a_{\kappa_t:t-1}\}$. Given the latent state $x_{\kappa_t}$ inferred from the former partition, we can express the belief over $x_t$, conditioned on both $x_{\kappa_t}$ and the delayed observations/actions, as shown in the second line. Finally, the third line shows how this belief can be recursively computed by applying the transition model $\psi$ over the latent states step-by-step, starting from $x_{\kappa_t}$ and incorporating the delayed information up to time $t$. For ease of notation, we have omitted the variables in $\psi$, but note that each has the form $\psi(x_\tau \mid x_{\tau-1}, \bar{o}_{\kappa_t+1:t}, a_{\tau-1})$.

Eq. (5) implements the exact Bayesian filter (Chen et al., 2003) for the available history. Assuming the learned models $p_\theta$ and $q_\theta$ match the true dynamics and posterior, the auxiliary kernel $\psi$ alternates between a prediction step (via $p_\theta$) and a measurement update (via $q_\theta$) whenever a delayed observation is received. This process reproduces the posterior that an out-of-sequence-measurement filter would obtain after re-ordering the data (Bar-Shalom, 2002). When all observations arrive without delay (i.e., $D = 0$), the update reduces to the standard variational inference procedure used in RSSM.

The belief state we compute here is a sufficient statistic for control with respect to the information actually available at time $t$ as no policy can exploit observations that have not yet arrived. Thus, conditioning the policy on $\phi_t$ is optimal for that information set (Kaelbling et al., 1998). In practice, we can approximate the belief distribution $\phi_t$ using particle filtering (Ma et al., 2020b) to capture uncertainty. Each step of the recursion in Eq. (5) is then represented with $K$ particles $\{x_t^{(k)}\}_{k=1}^K$, each propagated through the model according to the rule defined by $\psi$.

## 4.2 INCORPORATING INTO RL

We present a general training procedure for incorporating belief inference into MBRL under delayed observations. This framework, outlined in Algorithm 1, can be applied to any algorithm employing RSSM style world model. Modifications to the standard pipeline are highlighted in blue. The key idea is to train the world model on complete, ordered trajectories as in undelayed settings, while training the policy on belief states inferred from partially observed sequences using Eq. (5).

During inference the agent maintains a time-stamped buffer that stores observations that may arrive with delay. At every step $t$ the buffer is updated to contain only those observations whose indices belong to indices $\mathcal{J}_t$ indicating which observations have been received at time $t$. Therefore, the length of buffer can grow by maximum delay $D$, however, in practice one could set this maximum delay to some pre-defined constant that trades-off inference memory with performance. The latent dynamics hidden state is advanced to $\kappa_t$, the most recent time for which the observation sequence is complete. Starting from this latent state, the agent computes the current belief with the buffer and Eq. (5). The agent then selects an action from the policy defined on the belief space $\pi(\cdot \mid \phi_t)$.

While training happens in a delayed environment, world model training remains identical to the standard setting. This is possible because learning and data collection processes are decoupled. In

---

$^2$When $\mathcal{A}$ employs an observation decoder, each particle independently reconstruct the observation.

Table 1: Return of methods for different delay distributions across MuJoCo environments. Results are presented as the mean $\pm$ standard error of the mean over 5 seeds.

| Delay | Environment | DCAC | Encoding | Stack-Dreamer | DA-Dreamer |
|---|---|---|---|---|---|
| $\mathcal{U}\{0,10\}$ | HalfCheetah-v4 | $3841.43 \pm 455.59$ | $4179.97 \pm 198.08$ | $1959.96 \pm 428.36$ | $\mathbf{4985.40 \pm 129.13}$ |
| | Hopper-v4 | $\mathbf{2394.67 \pm 368.34}$ | $\mathbf{2389.69 \pm 139.21}$ | $\mathbf{2694.22 \pm 212.60}$ | $2251.36 \pm 211.03$ |
| | Humanoid-v4 | $1062.81 \pm 126.97$ | $559.22 \pm 16.77$ | $522.13 \pm 21.16$ | $\mathbf{1854.26 \pm 104.61}$ |
| | HumanoidStandup-v4 | $145293.09 \pm 2201.22$ | $113311.06 \pm 5654.87$ | $93154.06 \pm 9101.54$ | $\mathbf{220017.11 \pm 12077.36}$ |
| | Reacher-v4 | $\mathbf{-6.36 \pm 0.28}$ | $-7.05 \pm 0.10$ | $-6.81 \pm 0.12$ | $\mathbf{-6.37 \pm 0.07}$ |
| | Swimmer-v4 | $40.53 \pm 0.79$ | $121.54 \pm 12.18$ | $\mathbf{346.58 \pm 1.28}$ | $\mathbf{347.00 \pm 2.88}$ |
| $\mathcal{U}\{0,20\}$ | HalfCheetah-v4 | $2144.86 \pm 268.52$ | $\mathbf{4240.31 \pm 104.29}$ | $1045.15 \pm 91.71$ | $2958.56 \pm 27.65$ |
| | Hopper-v4 | $6.48 \pm 0.42$ | $1719.96 \pm 91.33$ | $\mathbf{2981.87 \pm 140.38}$ | $1713.85 \pm 175.05$ |
| | Humanoid-v4 | $112.08 \pm 39.25$ | $544.54 \pm 9.40$ | $418.24 \pm 21.51$ | $\mathbf{855.97 \pm 48.86}$ |
| | HumanoidStandup-v4 | $139806.95 \pm 10244.26$ | $118297.58 \pm 3845.44$ | $101241.22 \pm 2342.48$ | $\mathbf{195611.38 \pm 7214.55}$ |
| | Reacher-v4 | $\mathbf{-6.59 \pm 0.22}$ | $-7.31 \pm 0.15$ | $\mathbf{-6.71 \pm 0.11}$ | $-7.06 \pm 0.08$ |
| | Swimmer-v4 | $34.19 \pm 1.73$ | $117.39 \pm 10.32$ | $\mathbf{348.32 \pm 1.64}$ | $\mathbf{349.60 \pm 1.80}$ |

particular, after each episode terminates, we wait until all pending observations arrive before storing the ordered trajectory in the replay buffer. To make policy training consistent with deployment, i.e., delay-aware, the replay buffer is augmented with the indices $\mathcal{J}_t$. Replaying these indices allows us to reconstruct the same partial buffers, recompute beliefs, and provide them as inputs to the downstream policy learning algorithm.

## 5 EXPERIMENTS

We evaluate our method through two sets of experiments. Section 5.1 focuses on fully observable MuJoCo environments (Todorov et al., 2012), where we compare against MDP baselines. Since existing methods do not address our setting, we use methods designed for MDPs for a fair comparison within the scope of available baselines. Section 5.2 evaluates our method on four Meta-World environments (Yu et al., 2019) with visual inputs, which are inherently partially observable. In Meta-World, we compare against practical heuristics commonly adopted in the presence of delays, as they represent the simple strategies in the absence of delay-aware methods.

Our method builds on Dreamer-v3 (Hafner et al., 2025), an MBRL algorithm that learns a RSSM based world model, as described in Section 2.1. We modify Dreamer according to the procedure sketched in Algorithm 1 and use its default hyperparameters in all experiments. Most experiments use a single particle ($K = 1$) to approximate the belief for computational efficiency, as we observed no significant performance differences across different values of $K$ (see Appendix C.3). Dreamer's world model consists of both deterministic and stochastic paths, allowing it to maintain information over long horizons (Hafner et al., 2020; 2023) and making single particle sufficient. We refer to the version of Dreamer that treats received observations as generic inputs and stack them together as `Stack-Dreamer` (see Eq. (2)), and the version using **D**elay-**A**ware Algorithm 1 as `DA-Dreamer`. Each experiment is repeated for 5 different seeds.

### 5.1 MAIN COMPARISON WITH BASELINES

For MuJoCo environments, we compare against `DCAC` (Bouteiller et al., 2020) and the best-performing method of Wang et al. (2023), referred to as "detach Encoding" in their work, which we refer to simply as `Encoding` for clarity. We use the same architecture and hyperparameters as reported in their paper for consistency. Throughout, we denote the uniform distribution as $\mathcal{U}\{a, b\}$ and truncated Gaussian with $\mathcal{N}^+(\mu, \sigma^2)$.

### 5.1.1 RESULTS

Table 1 reports the returns of all methods under two uniform delay distributions. `DA-Dreamer` achieves better performance than other methods in more environments. In contrast, `Stack-Dreamer` performs well in simpler settings but fails to scale to environments with larger observation spaces, such as Humanoid and HumanoidStandup. This supports our earlier argument that treating delayed observations as generic inputs leads to an unnecessarily complex

Table 2: Return under different levels of stochasticity of the environment($\alpha$) in HalfCheetah-v4. Results are presented as the mean $\pm$ standard error of the mean.

| Delay | Environment | DCAC | Encoding | Stack-Dreamer | DA-Dreamer |
|---|---|---|---|---|---|
| | HalfCheetah-v4 ($\alpha = 0.2$) | $1952.64 \pm 609.37$ | $\mathbf{3361.24 \pm 163.53}$ | $1847.38 \pm 243.85$ | $\mathbf{3354.77 \pm 43.43}$ |
| | HalfCheetah-v4 ($\alpha = 0.4$) | $1735.57 \pm 56.46$ | $\mathbf{2594.15 \pm 112.69}$ | $947.06 \pm 178.85$ | $2362.95 \pm 32.64$ |
| $\mathcal{U}\{0, 10\}$ | HalfCheetah-v4 ($\alpha = 0.6$) | $1432.85 \pm 196.84$ | $\mathbf{1747.72 \pm 109.66}$ | $717.16 \pm 179.19$ | $\mathbf{1677.27 \pm 26.86}$ |
| | HalfCheetah-v4 ($\alpha = 0.8$) | $772.37 \pm 99.87$ | $865.24 \pm 25.91$ | $388.79 \pm 115.07$ | $\mathbf{1069.86 \pm 46.51}$ |
| | HalfCheetah-v4 ($\alpha = 1$) | $-26.84 \pm 67.14$ | $338.32 \pm 69.58$ | $72.25 \pm 15.11$ | $\mathbf{553.90 \pm 39.55}$ |

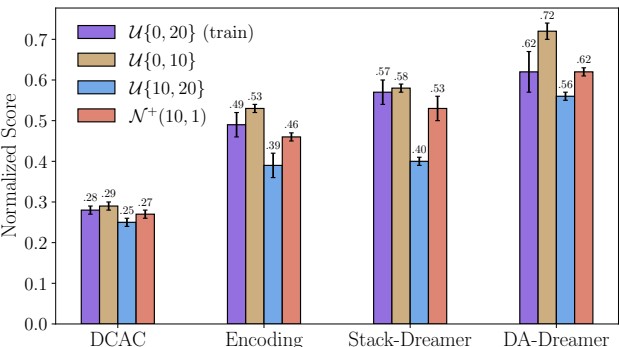

Figure 2: Normalized return under different test-time delay distributions. All methods are trained with $\mathcal{U}\{0, 20\}$ delays and evaluated on $\mathcal{U}\{0, 10\}$ (short delays), $\mathcal{N}^+(10, 1)$ (centered delays), and $\mathcal{U}\{10, 20\}$ (long delays). Bars and caps represent the mean and standard error of the mean.

latent space and fails to exploit the structure of delays. `DCAC` shows a sharp performance drop in Hopper and Humanoid under $\mathcal{U}\{0, 20\}$. These environments may terminate early due to unsafe joint configurations, exposing the limitations of augmentation-based methods under longer delays. `Encoding` demonstrates more stable performance but consistently underperforms `DA-Dreamer`. While the baselines rely on the MDP assumption and are not burdened by integrating past information, `DA-Dreamer` consistently outperforms them.

### 5.1.2 STOCHASTIC ENVIRONMENTS

While MuJoCo environments are deterministic apart from the initial state, we introduce added Gaussian noise with variance $\alpha$ into the normalized action space to evaluate the robustness of each method under stochastic environments. Table 2 reports the performance of all methods under delay distribution $\mathcal{U}\{0, 10\}$ in the HalfCheetah environment across different noise levels. `DA-Dreamer` and `Encoding` maintain competitive performance as noise increases, demonstrating greater robustness. In high-noise settings, `DA-Dreamer` achieves the best performance. `DCAC` shows high performance in the noise-free setting for this environment (Table 1), but its performance degrades sharply under high stochasticity. Similarly, `Stack-Dreamer` struggles even under mildly stochastic conditions. In such regimes, the agent must estimate a belief over latent states to act reliably. By explicitly computing this belief, `DA-Dreamer` effectively accounts for uncertainty, which is reflected in its strong performance.

### 5.1.3 EVALUATION ON UNSEEN DELAY DISTRIBUTION

In this section, we evaluate how well each method generalizes to delay distributions different from the one used during training. All methods are trained with delay distribution $\mathcal{U}\{0, 20\}$, and evaluated on three test distributions: $\mathcal{U}\{0, 10\}$ representing shorter delays, $\mathcal{N}^+(10, 1)$ showing the mean of training distribution, and $\mathcal{U}\{10, 20\}$ for longer ones.

Figure 2 reports the normalized return for each method, averaged across environments. We compute normalized return as $\nu(R) = \frac{R - R_{\min}}{R_{\max} - R_{\min}}$, with $R_{\min}$ and $R_{\max}$ being the return of random policy and Dreamer trained in undelayed environment, respectively. A well designed method to handle

Table 3: Success rate of methods for different delay distributions in Meta-World environments. Results are presented as the mean $\pm$ standard error of the mean.

| Delay | Environment | `Wait` | `Memoryless` | `DA-Dreamer` |
|---|---|---|---|---|
| $\mathcal{U}\{0, 5\}$ | button-press-wall-v2 | $0.0 \pm 0.0$ | $\mathbf{0.99 \pm 0.00}$ | $0.84 \pm 0.05$ |
| | drawer-close-v2 | $\mathbf{1.00 \pm 0.00}$ | $\mathbf{1.00 \pm 0.00}$ | $\mathbf{1.00 \pm 0.00}$ |
| | plate-slide-v2 | $0.0 \pm 0.0$ | $0.65 \pm 0.16$ | $\mathbf{0.97 \pm 0.03}$ |
| | reach-v2 | $0.64 \pm 0.03$ | $\mathbf{1.00 \pm 0.00}$ | $\mathbf{1.00 \pm 0.00}$ |
| $\mathcal{N}^+(10, 3)$ | button-press-wall-v2 ($\alpha = 0.25$) | $0.0 \pm 0.0$ | $\mathbf{0.45 \pm 0.05}$ | $\mathbf{0.48 \pm 0.14}$ |
| | drawer-close-v2 ($\alpha = 0.25$) | $0.30 \pm 0.04$ | $\mathbf{1.00 \pm 0.00}$ | $\mathbf{1.00 \pm 0.00}$ |
| | plate-slide-v2 ($\alpha = 0.25$) | $0.0 \pm 0.0$ | $\mathbf{0.43 \pm 0.06}$ | $\mathbf{0.49 \pm 0.09}$ |
| | reach-v2 ($\alpha = 0.25$) | $0.0 \pm 0.0$ | $0.78 \pm 0.02$ | $\mathbf{1.00 \pm 0.00}$ |
| $\mathcal{N}^+(20, 1)$ | button-press-wall-v2 | - | $0.10 \pm 0.06$ | $\mathbf{0.87 \pm 0.09}$ |
| | drawer-close-v2 | - | $\mathbf{1.00 \pm 0.00}$ | $\mathbf{1.00 \pm 0.00}$ |
| | plate-slide-v2 | - | $0.00 \pm 0.00$ | $\mathbf{0.70 \pm 0.13}$ |
| | reach-v2 | - | $0.25 \pm 0.07$ | $\mathbf{1.00 \pm 0.00}$ |

random delays should generally achieve improved performance under shorter delays. Otherwise, one could simply use a fixed-delay methods by setting the delay to a maximum possible value.

In `DCAC` and `Stack-Dreamer` minimal performance improvement on shorter delays is observed. `DCAC` performs poorly even under the training distribution, and its low sensitivity to test-time changes reflects limited impact on weak performance rather than robustness. Its reliance on fixed-size state augmentation also prevents its applicability to delays longer than those seen during training. `Stack-Dreamer` shows greater degradation under longer delays, highlighting the failure of treating observations as generic inputs to capture the temporal structure induced by random delays.

In contrast, `DA-Dreamer` generalizes well across all delay distributions, achieving much higher performance under shorter delays while remaining stable under longer ones. This is a desirable property, as delay distributions are often unknown in real-world settings, and methods trained on a distribution with wide support must remain reliable at deployment. `Encoding` exhibits a similar pattern, though smaller extent.

## 5.2 IMPORTANCE OF ADDRESSING DELAYS

In Meta-World, existing baselines are not applicable, so we used two alternatives commonly used in practice. The `Memoryless` agent uses the latest available observation in place of the current one. `Wait` method pauses to receive a new observation, where pausing is implemented by issuing no-op (zero) actions during waiting steps. While `Wait` is not always feasible in practice, we include it for comparison purposes. To evaluate whether methods can complete the task *in-time* while making delays more impactful, we reduce the default episode length in Meta-World environments to 50 steps, yet sufficient for the tasks considered.

### 5.2.1 RESULTS

Table 3 reports the final success rate, defined as the average number of successful episodes. The `Wait` agent is only effective in the simplest task under short delays; as it fails entirely in most tasks, we exclude it from evaluations under longer delays. `Memoryless` performs well under short delays but degrades quickly as delays increase, failing on more complex tasks. In contrast, `DA-Dreamer` consistently outperforms both baselines and maintains a high success rate even under long delays relative to the episode length. In stochastic environments, both `Memoryless` and `DA-Dreamer` experience performance drops on harder tasks, which is expected given the increased task complexity.

Additionally, we include experimental details and further results in the appendix, including training curves, ablations on `Stack-Dreamer` and `DA-Dreamer` , a study on the effect of number of particles, and visualizations of reconstructed observations.

## 6 RELATED WORK

Research on delayed RL began with foundational works that established unified frameworks for MDPs with both observation and action delays (Altman & Nain, 1992; Katsikopoulos & Engelbrecht, 2003). Early methods like dSARSA (Schuitema et al., 2010) used memoryless policies based on the last observation, but their performance quickly deteriorates with increasing delays. Augmentation-based methods construct extended states from the last observation and subsequent actions. For example, DC/AC proposed to resample trajectory fragments in hindsight (Bouteiller et al., 2020; Haarnoja et al., 2018), though these approaches suffer from the curse-of-dimensionality. Model-based strategies aim to infer the current state from extended states (Walsh et al., 2007; Derman et al., 2021) or to learn compact belief representations (Liotet et al., 2021). More recent works explore various model-based design heuristics in deep RL (Wang et al., 2023) or use world models to predict the state (Karamzade et al., 2024; Valensi et al., 2024).

Other works apply imitation learning from undelayed experts (Liotet et al., 2022) or reformulate delayed RL as variational inference solved with behavior cloning (Wu et al., 2024a), though these face policy mismatch (Liotet et al., 2022). Auxiliary-task methods (Wu et al., 2024b) introduce shorter delays to support training under long delays. BPQL (Kim et al., 2023) avoids full augmentation by projecting critic evaluations onto the original state space but struggles under high stochasticity.

Most prior methods assume constant delays. Random delays remain underexplored, with only a few approaches (Bouteiller et al., 2020; Wang et al., 2023; Valensi et al., 2024) addressing them, and only in fully observable MDPs without facing OOS observations. In the partially observable setting, Kim & Jeong (1987) studied lagged observations without proposing a learning method, while Karamzade et al. (2024) addressed constant delays without having OOS observations. Our work filled this gap by targeting stochastic observation delays in POMDPs.

Model-based reinforcement learning (MBRL) has demonstrated strong performance and superior sample efficiency compared to model-free methods (Hafner et al., 2019; Janner et al., 2019). Many state-of-the-art MBRL algorithms, such as Dreamer, SLAC, and STORM, rely on RSSMs (Hafner et al., 2020; Lee et al., 2020; Zhang et al., 2023), which represent latent dynamics through both deterministic and stochastic components. For belief estimation in POMDPs, several works have explored the use of Bayesian and particle filtering techniques (Igl et al., 2018; Ma et al., 2020a), enabling more accurate and uncertainty-aware latent state tracking in sequential environments.

## 7 CONCLUSION

We address the challenge of random observation delays in POMDPs by proposing a framework on top of model-based approaches, to effectively processes OOS observations for RL. Unlike prior methods, our approach does not rely on full observability or fixed delays, making it applicable to more realistic scenarios. Experiments on synthetic and simulated robotic environments show that our method outperforms baselines in MDP settings and remains effective under partial observability with random delays. It also generalizes well to unseen delay distributions, an essential feature for real-world applications.

A key limitation of our approach is the reliance on recursive filtering, which may accumulate one-step prediction errors over time and hinder scalability in long-horizon tasks. Future work could address this issue by exploring more scalable architectures, such as Transformer-based models or multi-step models. Moreover, we assumed that delays are finite and that no observations are permanently missing. While our framework can be extended to address missing observations by training the world model on subsequences of trajectories, we believe more effective approaches could be developed that are specifically designed for this setting.

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

## A  EXPERIMENTAL DETAILS

All methods and baselines were trained for $10^6$ environment steps. We used an action repeat of 2 for the Meta-World environments, resulting in $5 \times 10^5$ decision steps during training. For the baselines, we used the exact same hyperparameters as reported in their papers. For Dreamer-v3, we disabled the replay value loss, which prevents training the critic on data stored in the replay buffer; thus, the critic is only trained on generated trajectories during the policy learning phase. Additionally, we increased the number of classes in the discrete latent state representation to 32 and the number of neurons per layer to 512 for the Gym MuJoCo tasks only. In Meta-World, we used dense reward signals and $(64 \times 64)$ RGB images from `camera_id=1` as observations.

## B  TRAINING CURVES

Below, we include the training returns for the experiments presented in Table 5.1 and Table 5.2, respectively.

### B.1  GYM MUJOCO

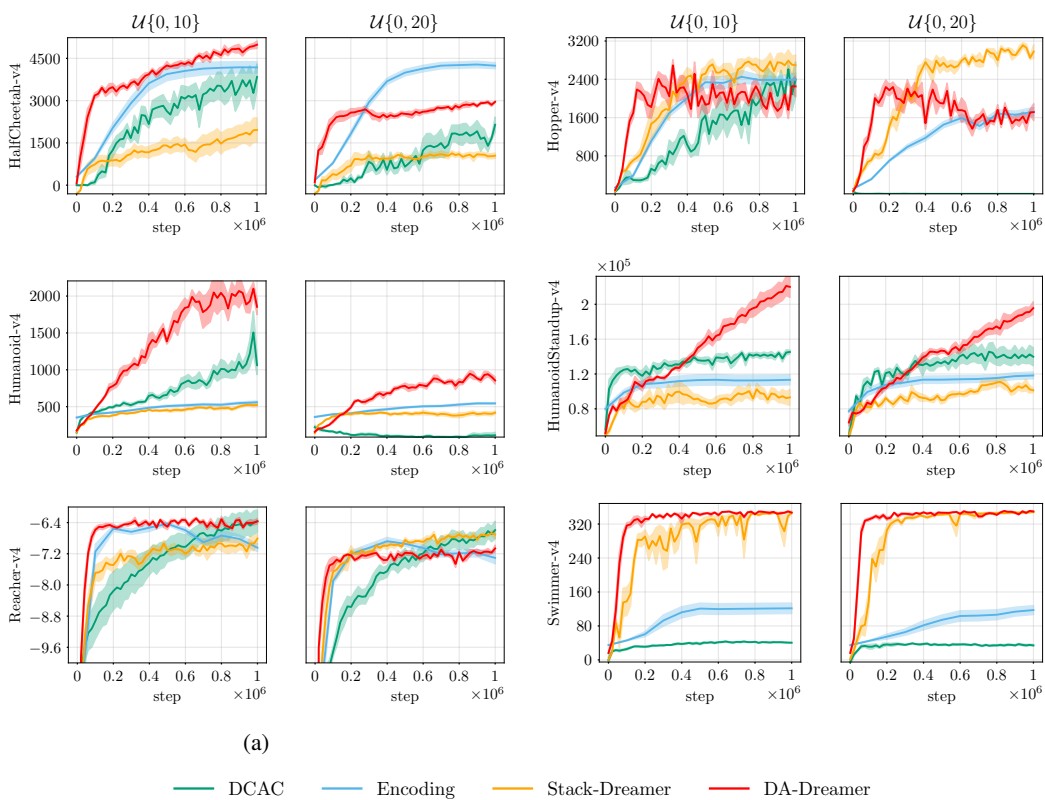

(a)

Figure 3: Learning curve of methods on selected Gym environments.

## B.2 META-WORLD

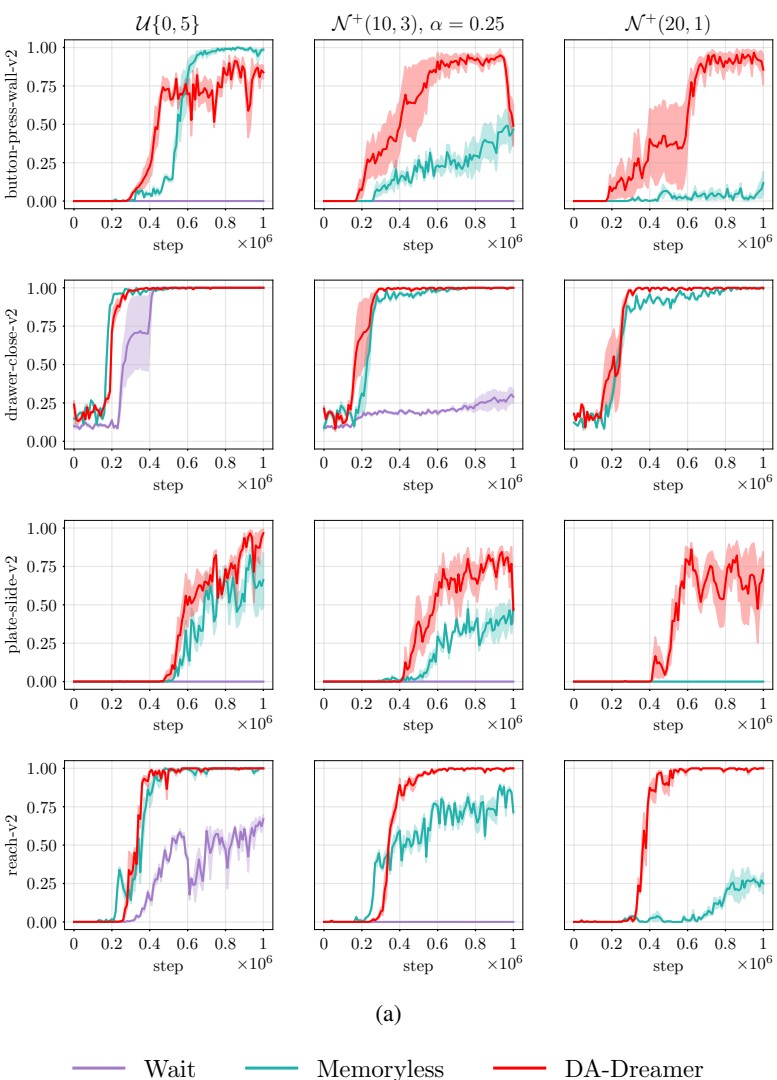

(a)

Wait — Memoryless — DA-Dreamer

Figure 4: Learning curve of methods on selected Meta-World environments.

## C ADDITIONAL EXPERIMENTS

### C.1 DELAY AWARE INFERENCE WITHOUT TRAINING

Table 4 shows the final return of `DA-Dreamer` when trained in a standard (non-delayed) environment but deployed in a delayed environment. Compared to training directly in the delayed setting, it underperforms in half of the environments, shows similar performance in two, and achieves higher scores in HumanoidStandup. We speculate, based on Figure 3, that the original method has not yet converged in HumanoidStandup, and with additional training, the performance gap may close. Overall, this approach appears promising for scenarios where the delay distribution is unknown during training, or when the agent must be trained once and deployed under varying, unknown delays.

Table 4: Return of the `DA-Dreamer` ablation on Gym environments. An asterisk (*) indicates similar performance.

| Delay | Environment | DA-Dreamer (inference only) |
|---|---|---|
| $\mathcal{U}\{0, 10\}$ | HalfCheetah-v4 | $2642.02 \pm 328.65$ |
| | Hopper-v4 | $1692.25 \pm 818.04$ |
| | Humanoid-v4 | $1808.95 \pm 322.17$ |
| | HumanoidStandup-v4 | $\mathbf{267659.35 \pm 26792.04}$ |
| | Reacher-v4 | $-6.13 \pm 0.12^*$ |
| | Swimmer-v4 | $349.99 \pm 1.29^*$ |
| $\mathcal{U}\{0, 20\}$ | HalfCheetah-v4 | $1384.41 \pm 119.14$ |
| | Hopper-v4 | $1430.39 \pm 865.04$ |
| | Humanoid-v4 | $722.37 \pm 134.74$ |
| | HumanoidStandup-v4 | $\mathbf{207792.76 \pm 12833.97}$ |
| | Reacher-v4 | $-7.08 \pm 0.14^*$ |
| | Swimmer-v4 | $350.35 \pm 1.75^*$ |

## C.2 STACKING OBSERVATIONS

In Figure 5, we experiment with different ways of stacking delayed information in `Stack-Dreamer`. The default version, reported in the main text, inputs the previous $D$ observations and actions, with missing observations filled with zeros. We also evaluate a variant that removes actions from the input, as they are already represented in the latent state from the previous time step, and another variant that additionally includes a mask indicating which observations have been received. As shown, all variants perform similarly, with no significant differences observed between them.

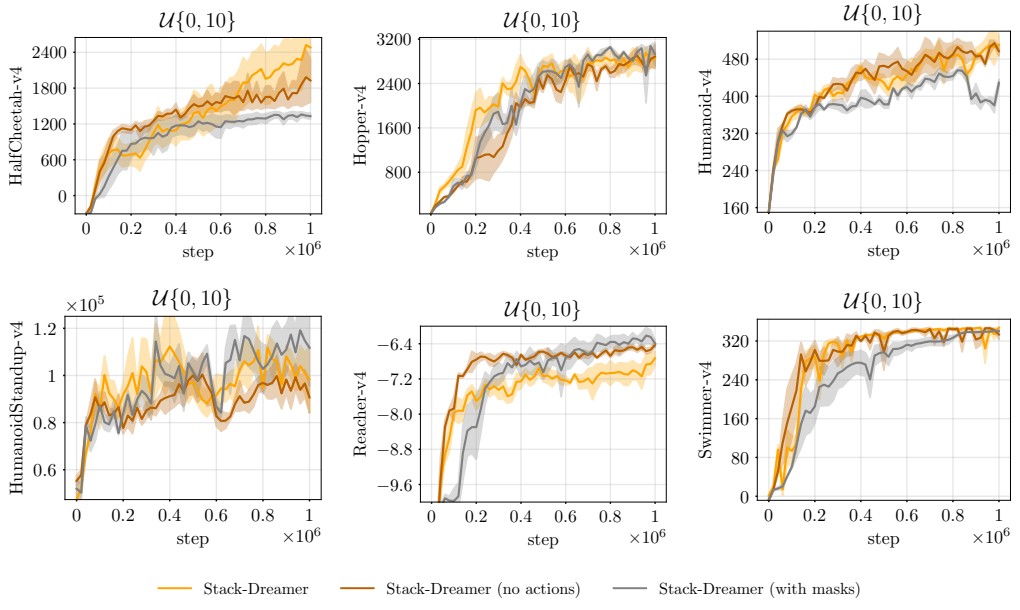

Figure 5: Learning curve of `Stack-Dreamer` variants.

## C.3 NUMBER OF PARTICLES

Figure 6 shows the performance of `DA-Dreamer` with different numbers of particles $K$. In our implementation, we simply concatenate the $K$ particles, though various alternatives exist for combining them (Ma et al., 2020a). As shown, increasing the number of particles generally improves performance, but the gains are not substantial. We hypothesize that this is because the Dreamer-v3

world model includes both deterministic and stochastic components, with the deterministic part capable of retaining information along a trajectory. To balance performance with computational cost, we used $K = 1$ in the main experiments.

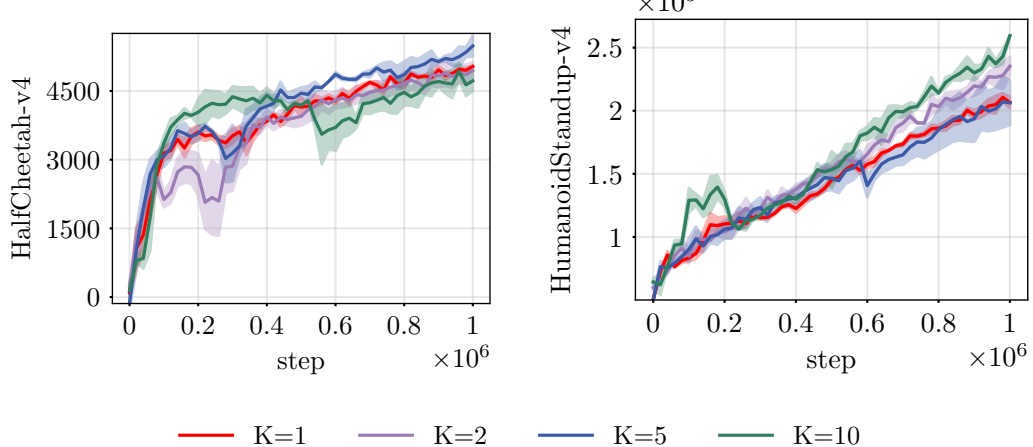

Figure 6: Performance vs different number of particles.

### C.4 VISUALIZING RECONSTRUCTED FRAMES

Figure 7 shows reconstructed observations from delayed information in `DA-Dreamer`. As seen, for shorter delays (a), the reconstructed frames are sharper compared to longer delays (b). This is expected, as longer delays increase uncertainty in the belief state about the current environment state. Consequently, the belief assigns more weight to nearby states, resulting in blurrier reconstructed frames, as depicted.

## D CODE RELEASE

We have uploaded the code, which modifies Dreamer-v3 based on our method. The upload includes example scripts for running the experiments. We plan to make the code publicly available in a GitHub repository upon acceptance.

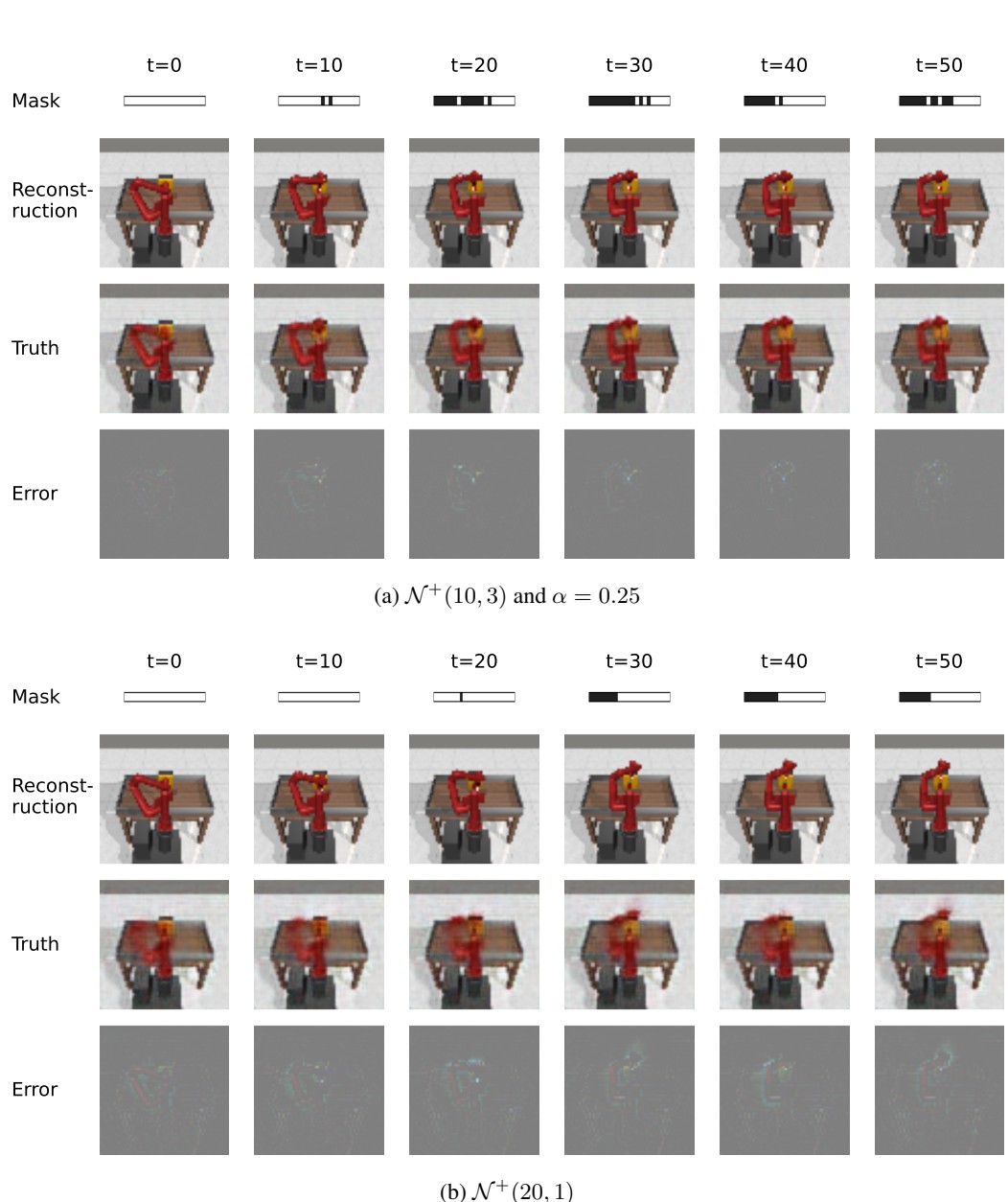

(a) $\mathcal{N}^+(10, 3)$ and $\alpha = 0.25$

(b) $\mathcal{N}^+(20, 1)$

Figure 7: Reconstructing the current observation in delayed button-press-v2 from the computed belief state. The mask indicates the arrival of past observations (black cells denote received observations), with the rightmost cell representing the current timestep.

