# OpenReview forum: "Model-Based Reinforcement Learning under Random Observation Delays"
_ICLR.cc/2026/Conference — ICLR 2026 Conference Withdrawn Submission_

### Official Review · Reviewer_RtcN · 2025-10-26

**Soundness:** 3
**Presentation:** 2
**Contribution:** 1
**Rating:** 4
**Confidence:** 5

**Summary:**

In this paper, authors propose a novel model-based delayed RL framework targeting the POMDP setting. Specifically, author leverage Dreamer RSSM framework to sequentially update the belief state and handle random delays.

**Strengths:**

The paper overall is well-written with clear structure and comprehensive experiments. Reviewer particularly appreciate authors' effort to elaborate delay setting under more applied robotics scenario and extend experiments into more robotic manipulation tasks comparing with current delay adaption in the robotics field.

**Weaknesses:**

1, There are bunch of model-based delayed RL methods [1,2,3], and the delayed MDP in nature is a POMDP problem. Particularly, Dreamer structure has already been applied in the following work[3] for the online delayed RL problem. Thus, it would be better if author can clearly illustrate the distinction, novelty, formulation, and contribution comparing with existing literature in the introduction.\
2, Comparison with current SOTA methods are missing. For model-free, authors might consider to add comparison with either [4] or [5]. For model-based, author might consider to compare with transformer based methods such as [1,2].\
3, The resolution of robotic task in appendix are very unclear. Please update a high-solution version.\
4, It would be better to incorporate larger delay steps in the environment setup. Since reviewer concerns that the prediction power of current Latent-state space and ensemble MLP models might deteriorate as delay step grows.\
5, The author might want to illustrate more on the RL part in section 4.2. Is the training conducted on the predicted observation or augmented history. Besides, aside from the observation augmentation, what's distinction between this work with the original Dreamer paper?

[1]:Wu, Qingyuan, et al. "Directly forecasting belief for reinforcement learning with delays." arXiv preprint arXiv:2505.00546 (2025).\
[2]:Liotet, Pierre, Erick Venneri, and Marcello Restelli. "Learning a belief representation for delayed reinforcement learning." 2021 International Joint Conference on Neural Networks (IJCNN). IEEE, 2021.\
[3]:Karamzade, Armin, et al. "Reinforcement learning from delayed observations via world models." arXiv preprint arXiv:2403.12309 (2024).\
[4]:Kim, J., Kim, H., Kang, J., Baek, J., and Han, S. Belief projection-based reinforcement learning for environments with delayed feedback. In Thirty-seventh Conference on Neural Information Processing Systems, 2023.\
[5]:Wu, Q., Zhan, S. S., Wang, Y., Wang, Y., Lin, C.-W., Lv, C., Zhu, Q., and Huang, C. Variational delayed policy optimization. Advances in neural information processing systems, 2024a.

**Questions:**

See above.

---

### Official Review · Reviewer_HXPd · 2025-10-27

**Soundness:** 2
**Presentation:** 2
**Contribution:** 2
**Rating:** 4
**Confidence:** 4

**Summary:**

This paper proposes DA-Dreamer, advancing reinforcement learning under random observation delays.
Specifically, DA-Dreamer proposes a model-based filtering process to estimate the observations arriving out-of-sequence.
Experimental results on various benchmarks demonstrate that DA-Dreamer achieve superior performance and robustness compared to existing baselines.

**Strengths:**

1. This paper investigates an interesting problem setting, out-of-sequence and random observation delays.
2. Experimental results show that DA-Dreamer outperforms DCAC and other baselines.

**Weaknesses:**

1. Some important model-based delayed RL baselines [1-4] are missed. It would be great to compare these (belief) model-based methods and highlight the novelty and contribution of this paper.
2. Can you compare the prediction accuracy of the proposed belief architecture?
3. As mentioned by the authors, the limitation of DA-Dreamer is that it uses recursive filtering, resulting in compounding errors. And this issue can be addressed by sequence modelling methods [4] (e.g., transformer or diffusion). It would be great to incorporate the proposed method into the transformer or diffusion.


>Reference
>>
>> [1] Delay-aware model-based reinforcement learning for continuous control
>>
>> [2] Reinforcement learning from delayed observations via world models
>>
>> [3] Learning a belief representation for delayed reinforcement learning
>>
>> [4] Directly Forecasting Belief for Reinforcement Learning with Delays

**Questions:**

See weakness.

---

### Official Review · Reviewer_t8Ay · 2025-10-29

**Soundness:** 2
**Presentation:** 2
**Contribution:** 2
**Rating:** 4
**Confidence:** 4

**Summary:**

This paper studies model-based reinforcement learning (MBRL) under random observation delays, a setting where sensor data arrives out of sequence. The authors formalize this as a POMDP with stochastic observation latency and propose a “Delay-Aware Dreamer” (DA-Dreamer) framework that integrates a latent-space Bayesian filter into a Dreamer-like architecture. The method updates beliefs when delayed observations arrive and achieves improved robustness across several MuJoCo and Meta-World tasks with random delays.

**Strengths:**

- The paper provides a well-structured formalization of random observation delay as an augmented POMDP and connects it to Bayesian filtering.

- Empirical validation: The experiments cover multiple delay distributions and demonstrate strong performance over MDP-based baselines.

**Weaknesses:**

- Once each observation arrives with its timestamp, the realized delay is known. In that case, belief estimation reduces to sequence modeling over timestamped observations, which is a setting already handled by existing belief-based or transformer architectures [1]. [1] demonstrates that a transformer can predict the latent belief effectively in this exact situation. The present paper does not compare against such methods, which undermines its claim of novelty.

- Beyond integrating the filter into Dreamer, the paper does not provide new theory, analysis, or computational insight. There is no examination of convergence, computational complexity, or belief approximation error due to delays.



[1] Wu, Qingyuan, et al. "Directly Forecasting Belief for Reinforcement Learning with Delays." Forty-second International Conference on Machine Learning.

**Questions:**

How does DA-Dreamer differ conceptually from a transformer-based belief predictor that conditions on timestamped observations? In addition, is the proposed Bayesian recursion offering measurable advantages in efficiency, stability, or interpretability over learned belief forecasting?

---

### Official Review · Reviewer_jNiv · 2025-11-03

**Soundness:** 3
**Presentation:** 3
**Contribution:** 3
**Rating:** 6
**Confidence:** 3

**Summary:**

This paper studies random observation delays in POMDPs, where observations may arrive out-of-sequence (OOS). The authors formulate a random-delay POMDP problem, then proposes a latent-space belief filtering approach using a world model (RSSM), alternating between dynamics rollouts and posterior updates depending on whether delayed observations are available. The authors evaluate their approach on MuJoCo (fully observable) and Meta-World (partial-observability with vision), demonstrating strong performance vs. DCAC/Encoding/Stack-Dreamer and heuristics.

**Strengths:**

1. The authors study an important and novel problem. Prior delayed RL always assumes fix/constant delays while the random-delayed RL is underexplored.
2. The approach is clear as belief-filtering formulation with auxiliary kernel, which can be easily integrated into Dreamer , the model -based RL framework.
3. The experimental results look promising
4. The paper overall is well-written.

**Weaknesses:**

1. Baselines are not clearly explained
The paper does not clearly explain how the baseline methods actually work or how they differ from DA-Dreamer. It would help if the authors described what each baseline assumes about delays, how each handles out-of-order observations, and why those choices are limited. Then the experimental results would be easier to interpret, because readers could see how the differences in method lead to the differences in performance.

2. Unclear impact of model errors on belief filtering
DA-Dreamer depends on having an accurate learned world model. If the model is not perfect, the belief updates could accumulate error over time. It is not very clear in the paper how much this matters in practice or how sensitive the method is to model inaccuracies. More explanation or experiments would help show how robust the method is when the world model is not fully accurate.

3. Bounded delays and missing comparison to prior belief-based approaches
The paper assumes there is a maximum possible delay. However, prior belief-based delay RL methods could also handle random delays by always planning for the maximum delay. The authors do not compare against this, or explain why such a comparison might not be fair or meaningful. Including this discussion or experiment would make the contribution clearer and stronger.
Reference:
Wu, Qingyuan, et al. "Variational delayed policy optimization." Advances in neural information processing systems 37 (2024): 54330-54356.

**Questions:**

Please check the weaknesses.

---

### Note · Authors · 2025-11-13

I have read and agree with the venue's withdrawal policy on behalf of myself and my co-authors.